# Performance Degradation Assessment of Railway Axle Box Bearing Based on Combination of Denoising Features and Time Series Information

**DOI:** 10.3390/s23135910

**Published:** 2023-06-26

**Authors:** Zhigang Liu, Long Zhang, Qian Xiao, Hao Huang, Guoliang Xiong

**Affiliations:** 1Key Laboratory of Modern Transportation and Logistics of Jiangxi Province, Jiangxi Vocational and Technical College of Communications, Nanchang 330013, China; zhigangliu@ecjtu.edu.cn (Z.L.);; 2State Key Laboratory of Performance Monitoring Protecting of Rail Transit Infrastructure, East China Jiaotong University, Nanchang 330013, China

**Keywords:** performance degradation assessment, feature extraction, DRSN, LSTM, multi-domain features

## Abstract

In the existing rolling bearing performance degradation assessment methods, the input signal is usually mixed with a large amount of noise and is easily disturbed by the transfer path. The time information is usually ignored when the model processes the input signal, which affects the effect of bearing performance degradation assessment. To solve the above problems, an end-to-end performance degradation assessment model of railway axle box bearing based on a deep residual shrinkage network and a deep long short-term memory network (DRSN-LSTM) is proposed. The proposed model uses DRSN to extract local abstract features from the signal and denoises the signal to obtain the denoised feature vector, then uses deep LSTM to extract the time-series information of the signal. The healthy time-series signal of the rolling bearing is input into the DRSN-LSTM reconstruction model for training. Time-domain, frequency-domain, and time–frequency-domain features are extracted from the signal both before and after reconstruction to form a multi-domain features vector. The mean square error of the two feature vectors is used as the degradation indicator to implement the performance degradation assessment. Artificially induced defects and rolling bearings life accelerated fatigue test data verify that the proposed model is more sensitive to early failures than mathematical models, shallow networks or other deep learning models. The result is similar to the development trend of bearing failures.

## 1. Introduction

Rolling bearings are key components of mechanical rotating equipment, and their operating condition can determine the overall equipment performance life [1]. The effective information of rolling bearings operating conditions is contained in their vibration signal. The analysis of the vibration signal is helpful in understanding the status information of the bearings and to predicting their remaining life, which makes conditions for maintenance appropriate.

In general, bearings in mechanical equipment will undergo a performance degradation from normal to degradation and failure. Monitoring the operating status of bearings and analyzing their vibration signals can effectively detect early, weak faults and reveal their performance degradation trends, which can be utilized to develop focused equipment maintenance strategies. Therefore, the performance degradation assessment of rolling bearings is the premise and key point of implementing condition-based maintenance [2].

The bearing performance degradation assessment process can be divided into: Feature extraction, model establishment, and assessment analysis with indicators construction. In these three steps, a lot of research work has been done by domestic and foreign scholars. Liu et al. [3] combined Bayesian inference and self-organizing mapping to assess the performance degradation of rolling bearings. They used independent component analysis as the signal feature extraction method and selected a self-organizing network as the performance degradation assessment model, then constructed the quantitative bearing performance assessment index with negative log-likelihood probability, and finally calculated the bearing failure probability with Bayesian inference to implement early failure identification and performance degradation assessment of bearings. There is a problem that the traditional performance degradation indicator cannot accurately portray the performance degradation state of rolling bearings during the whole life cycle. Yang et al. [4] implemented the performance degradation assessment of rolling bearings using feature fusion and the gray regression method, which first extracted the time-domain features, energy features, and entropy features from the bearing vibration signal, then implemented feature screening and feature fusion on the constituent feature vectors, and finally used the gray regression model to construct performance degradation indicator. Two sets of rolling bearing full life cycle data verified the effectiveness of the proposed method. Zhang et al. [5] extracted useful features from gear vibration signals using the Autoregressive Model (AR), then trained an AANN model with AR coefficients of normal signals as feature vectors, used AR coefficients of signals to be measured as input vectors of trained AANN, outputted AR coefficient reconstruction vectors from the model, and constructed a fault assessment indicator using the root mean square error in the residual vectors of the AR model before and after reconstruction to achieve performance degradation assessment of gears. Lei et al. [6] proposed a performance degradation assessment model based on PCA-FCMAC (Fuzzy Cerebellar Model Articulation Controller), which first used PCA to downscale the signal features to eliminate redundant information, and then the downscaled features were input into FCMAC. The results verified the effectiveness and superiority of PCA-FCMAC.

Although good results in equipment performance degradation assessment can be achieved with the above methods, there are still some problems: (1) A series of feature engineering of signals has to be implemented such as feature selection, feature downscaling and feature fusion in the feature extraction process. Feature engineering requires a lot of effort and time in designing the preprocessing process and data conversion, which requires certain expert knowledge to judge the quality of feature extraction; (2) signals collected from sensors are inevitably mixed with a large amount of noise and transmission path interference, and the direct feature extraction of signals with noise and interference will influence the effect of performance degradation assessment; (3) most of the performance degradation assessment models are simple linear mathematical models or shallow networks, which have limited the ability to learn and extract features, and the volume of data that are input into model is quite low, which cannot meet the data analysis requirements in the background of big data; (4) the timing information of the bearing signal is usually ignored after inputting into the model. Especially for the performance degradation model of signal reconstruction [7], the timing information of the signal is quite important, ignoring the timing information of the signal will not be able to reconstruct the maximum information of the original signal, which will eventually influence the effect of the assessment results.

In recent years, with the rise of artificial intelligence, deep learning has been widely used in the field of equipment fault diagnosis [8]. The most representative deep learning models are Convolution Neural Network (CNN) [9,10], Recurrent Neural Network (RNN) [11,12], and Auto-Encoder (AE) [13,14]. The advantages of deep learning models are as follows: (1) Deep learning models can implement an “end-to-end” fault diagnosis [15], avoiding the tedious feature engineering in traditional methods; (2) deep learning models are complex nonlinear deep networks, and model training generally needs a large amount of data, which can meet the requirement of the big data background in mechanical equipment; (3) in CNN, the operation of convolution using convolution kernel can be understood as the process of denoising while extracting the features of the signal; (4) RNN can retain the temporal information of the data when processing data.

Considering the above advantages, a few scholars have started to use deep learning methods for equipment state performance assessment. Xu et al. [16] proposed a median filtered deep confidence network for constructing performance degradation indicators. Although the method has a good denoising effect on the signal and can obviously portray the bearing degradation process, the degradation indicator value exceeds the alarm threshold line only at the late stage of the degradation process, which is not conducive to the early bearing fault detection. Chen et al. [17] directly input the collected signals into a model consisting of CNN and RNN to construct indicators, which achieved good results. The problem of the proposed method is that the indicator labels have to be constructed first. The degradation trend of each part of each piece of equipment is different under actual working conditions, so the method they proposed has some limitations. Zhang et al. [18] implemented an “end-to-end” bearing performance degradation assessment using CNN and LSTM, which first trained models with normal healthy data, input test data into the trained models, and finally used the H-statistic to measure the degree of performance degradation, but the experimental data showed that the noise component in the bearing performance degradation curve was too much to accurately reflect the performance degradation process.

In summary, an end-to-end rolling bearing performance degradation assessment model based on DRSN-LSTM is proposed in this paper, in which DRSN is used to extract local abstract features and implement denoising operation from signals to obtain the denoising feature vector. The signal timing information is extracted using deep LSTM and then the original signal is reconstructed using two nonlinear layers. In order to comprehensively encompass the state information of the rolling bearing signal, the time-domain, frequency domain, and time-frequency domain features before and after signal reconstruction are extracted to form a multi-domain feature vector. Finally, the mean square error of two feature vectors is used as the degradation indicator, and the effectiveness of the proposed method for detecting early bearing faults and portraying the monotonicity and consistency of the performance degradation process is confirmed using the bearing fatigue test data.

## 2. The Framework of DRSN

The Deep Residual Shrinkage Network (DRSN) [19], is improved from the Deep Residual Network (ResNet), which is similar to ResNet in structure. The DRSN in this paper consists of a convolutional layer (Conv), a batch normalization (BN), an activation function, and a residual shrinkage building unit (RSBU).

### 2.1. Convolutional Layer

In the convolution layer, the convolution kernel implements the convolution operation with the input data to obtain the feature map. The convolution layer can be used to extract the required features by changing the parameters of the convolution kernel. The convolution operation of the convolution layer is shown in Equation (1). Different features of the data can be learned by using different convolution kernels during feature extraction. The formula for calculating the feature map output during the convolution operation is shown in Equation (2).
(1)$$y^{l(i,j)}=\sum_{j^{\prime }=0}^{k-1}\left(w_{i}^{l(j^{\prime })}x^{l(j+j^{\prime })}+b_{i}^{l(j^{\prime })}\right)$$
where $w_{i}^{l(j^{\prime })}$ is the *j*′-th weight of the *i*-th convolution kernel in the *l*-th layer, $b_{i}^{l(j^{\prime })}$ is the *j*’-th bias of the *i*-th convolution kernel in the *l*-th layer. The local field of perception of the *j*-th convolution kernel in the *l*-th layer is $x^{l(j+j^{\prime })}$ and *k* is the width of the convolution kernel.
(2)$$O=\frac{I-K+2P}{S}+1$$
where *I* is the input feature size. *O* is the output feature size. *K* is the convolutional kernel size. *S* is the convolutional step size of the convolutional kernel. *P* is the size of padding.

### 2.2. Activation Function

Since most of the problems, in reality, are nonlinear. In nonlinear problems, errors inevitably arise if modeled with linear functions. To enhance the nonlinear modeling capability, nonlinear activation functions have been introduced to CNN. The activation function layer enables CNN to learn complex and nonlinear features from the data. The activation function layer is generally placed between the convolutional layer and the pooling layer. The commonly used activation functions are the sigmoid function, the tanh function, and the linear rectification function (ReLU). ReLU is used as the activation function in DRSN.

The mathematical formula of ReLU is:(3)$$f(x)=\max\left\{0,x\right\}$$

### 2.3. Batch Normalization Layer

BN can change the distribution of feature value in the layer, pulling it back to the standard normal distribution. The feature value will be distributed in an interval where the activation function is more sensitive to the input. A small change in the input can correspond to a larger fluctuation in the loss function, resulting in a larger gradient, avoiding the problem of gradient disappearance and speeding up the training convergence. The formula of BN can be expressed as:(4)$$\mu =\frac{1}{N_{\mathrm{batch}\;}}\sum_{n=1}^{N_{\mathrm{batch}\;}}x_{n}$$
(5)$$\sigma^{2}=\frac{1}{N_{\mathrm{batch}\;}}\sum_{n=1}^{N_{\mathrm{batch}\;}}{\left(x_{n}-\mu \right)}^{2}$$
(6)$${\hat{x}}_{n}=\frac{x_{n}-\mu }{\sqrt{\sigma^{2}+\varepsilon }}$$
(7)$$y_{n}=\gamma {\hat{x}}_{n}+\beta$$
where $N_{\mathrm{batch}\;}$ represents the size of the batch, $x_{n}$ represents the input of the nth batch, $y_{n}$ represents the output of the nth batch, ε is a constant value close to zero, γ is a parameter value to measure the distribution, β is a parameter value to move the location of the distribution. Both γ and β can be obtained by training.

### 2.4. Residual Shrinkage Building Unit

In signal processing, the soft-threshold segmentation algorithm implements signal denoising by converting the near-zero feature values of the signal to zero. The soft-threshold function can be expressed as:(8)$$y=\left\{\begin{array}{cc} x-\tau & x>\tau \\ 0 & -\tau <x<\tau \\ x+\tau & x<-\tau \end{array}\right\}$$
where *x* is input, *y* is output, τ represents threshold value.

DRSN implements feature extraction with simultaneous denoising by introducing a soft thresholding function to construct RSBU which is the most important part of DRSN. The detailed structure of the RSBU is shown in Figure 1. C is the size of the feature map channel, W is the feature map width, and the height of the feature map is one. An RSBU consists of two BN, two ReLU, two convolutional layers, a thresholding module, a soft threshold setting module, and an identity shortcut. The feature map x after going through the absolute value layer in the threshold module, GAP layer, BN, ReLU, and two fully connected (FC) layers in the threshold module, the size of the features all became C. The output of the fully connected layer is input to Sigmoid layer. The output $\alpha_{c}$ is used as the scale parameter. There is the following formula $\alpha_{c}$:(9)$$\alpha_{c}=\frac{1}{1+e^{-z_{c}}}$$
where *Z_c_* is the node feature in channel C. $\alpha_{c}$ is multiplied with the absolute value layer and GAP layer to obtain the soft threshold value which is calculated as:(10)$$\tau_{c}=\alpha_{c}\cdot \;\mathrm{average}\;\left|x_{i,j,c}\right|$$

In Input $x_{i,j,c}$, *i*, *j*, and *c* represent the width, length, and the number of channels of the feature map *x*, respectively; $\tau_{c}$ is the threshold values of the input feature map *x*.

From the above, it can be seen that the threshold module can automatically set the threshold value during the training process of the network.

## 3. The Framework of LSTM

The traditional Deep Neural Networks (DNN) have the advantage of nonlinear data processing, but they are not sensitive to processing time-series data problems. The Recurrent Neural Network (RNN) can process time-series data through chain interconnection between its own neurons.

LSTM [20] introduces cellular memory units in the base of the hidden layer of RNN, which allows the model to preserve and control short-term state memory and effectively solves the problem of gradient disappearance or gradient explosion that exists in RNN.

In the training and testing of LSTM, the value of each hidden layer is determined using the current moment input and the previous moment hidden layer value. The memory unit structure of LSTM consists of three gate controllers, which are the input gate, the forgetting gate and the output gate. Three gated recurrent memory units implement the forgetting and updating of the memory state information, during which the key information is ensured to be retained by memory and the minor information is forgotten, ensuring the flow and storage of memory state information among hidden layers. The process keeps recurrent until the input of all temporal data is completed. The memory cell structure of LSTM is shown in Figure 2. The steps of the LSTM single hidden layer computation update state are:

Immediate memory state information of the current moment ${\hat{c}}_{t}$:(11)$${\hat{c}}_{t}=\tanh(W_{xc}x_{t}+W_{hc}h_{t-1}+b_{c})$$
forgotten gate $f_{t}$:(12)$$f_{t}=\sigma (W_{xf}x_{t}+W_{hf}h_{t-1}+b_{f})$$
input gate $i_{t}$:(13)$$i_{t}=\sigma (W_{xi}x_{t}+W_{hi}h_{t-1}+b_{i})$$

Current moment memory state information $c_{t}$:(14)$$c_{t}=c_{t-1}⊗f_{t}+\hat{c}⊗i_{t}$$
where ⊗ is the point-by-point product.

Output gate $o_{t}$:(15)$$o_{t}=\sigma (W_{xo}x_{t}+W_{ho}h_{t-1}+b_{o})$$

Output of the current hidden layer of the LSTM $h_{t}$:(16)$$h_{t}=o_{t}\tanh\left(c_{t}\right)$$

In Equations (11)–(16), $W_{xc}$, $W_{xi}$, $W_{xf}$ and $W_{xo}$ are the connection weight matrices between the input and hidden layers at time *t*, respectively; $W_{hc}$, $W_{hi}$, $W_{hf}$ and $W_{ho}$ are the connection weight matrices between the hidden layers at times *t* − 1 and *t*, respectively; $b_{c}$, $b_{i}$, $b_{f}$, $b_{o}$ are the biases of the input nodes, input gates, forgetting gates and output gates, respectively; σ represents the sigmoid functions.

## 4. The PDA Model Combining Denoising Features and Time Series Information

The flow of the proposed performance degradation assessment method is shown in Figure 3, which can be divided into a model training part and a model testing part. In the following, the structure of the whole model and its parameters are introduced first, and the process and parameter settings of model training and testing are described next.

### 4.1. Model Structure and Its Parameter

This model is a data reconstruction model. The encoder part consists of 20 DRSNs and 1 LSTM. The decoder consists of two fully connected layers, and the structural parameters of the DRSN-LSTM reconstruction model are shown in Table 1. Among them, the structure of the DRSN model used in this paper is similar to ResNet 18 in deep residual networks, which can be understood as adding a shrinkage unit to ResNet 18. The proposed network structure mainly consists of one convolutional layer, a batch normalization, an activation function layer and four blocks, where one block consists of two identical RSBU, one RSBU consists of a BN, a ReLU, one convolutional layer, a threshold module, a soft threshold setting module and an identity shortcut. The size of the convolution kernel of the RSBU in each of the four blocks is 3 × 1. The convolution stride in each of the two RSBU in Block1 is one, so the total stride size is one. The stride size of the first convolution layer in the first RSBU in Block2, Block3, and Block4 is two. The stride size of the second convolution layer is one. The convolution stride of two convolution layers in the other RSBU are both one. The total stride size of Block2, Block3 and Block4 are all two. LSTM input vector length is L/8, output vector length is L/32, and the number of layers is four.

### 4.2. Model Training Process

In this paper, the training process of the DRSN-LSTM data reconstruction model is to first randomly select data from the bearing health state signal dataset as model input, and intercept the signal into segmentation. This paper divides the signal into 20 segments. Each segment of the signal is denoised and downscaled separately using DRSN, which can encode a 20 × L signal vector into a 20 × L/8 temporal feature vector. In order to retain the spatial and temporal information of the vector, the 20 × L/8 timing feature vector is encoded using LSTM as 20 × L/32. The encoded vectors are flattened and input to two linear layers for signal reconstruction, and setting loss function to optimize the distance between the reconstructed signal $\hat{x}$ and input signal x as shown in Equation (17), where M is the length of the vector x. Finally, the number of iterations is set to ensure that the error value of the distance between $\hat{x}$ and x is limited to a certain range, and the network model trained using the health state signal is output.
(17)$$L_{\;}=\min\frac{1}{M}\sum {\left(x-\hat{x}\right)}^{2}$$

### 4.3. Model Testing Process

In the testing process, similar to training, data are selected one by one from the testing dataset as the input of the trained data reconstruction model. DRSN and LSTM encode input signals to obtain multiple sets of low-dimensional coding vectors that retain temporal information, and the flattened coding vectors are decoded with two-layer linear model to obtain the reconstructed signals. In order to fully obtain the information of signal features, the input signal and the reconstructed signal are extracted with multi-domain features, including 12 groups of time-domain features, such as mean value, standard deviation, square root amplitude, root mean square value, cliffs, peak value, margin, waveform factor and pulse indicator; 12 groups of frequency-domain features, such as spectral mean value, spectral root mean square value, frequency center of gravity, average energy, spectral concentration, main band position change, root mean square frequency and standard deviation frequency. The signal is decomposed with a three-layer db5 wavelet packet to obtain eight sets of time-frequency domain features with eight frequency sub-band energy ratios. Time-domain features, frequency-domain features, and time–frequency-domain features form a 32-dimensional input signal feature extraction vector z and a 32-dimensional reconstructed signal feature extraction vector $\hat{z}$. Based on the above discussion, it can be seen that the model is trained in a training dataset so that the model parameters can only learn the features and data distribution of bearing with a normal health state. When the test input is normal, health state data, multi-domain feature extraction vectors z and $\hat{z}$ have a high similarity. However, if the bearing failure occurs, it will be reflected in the vibration signal. So, when the test data are input to the network, the difference between the feature extraction vectors z and $\hat{z}$ will become larger. The mean square error (MSE) (Equation (18)) is used as the error measure of the feature extraction vectors for performance degradation assessment of rolling bearings.
(18)$$L_{\;}=\min\frac{1}{m}\sum {\left(z-\hat{z}\right)}^{2}$$
where *m* is the vector length of the multi-domain feature extraction vector z.

## 5. Analysis of Axle Box Bearing Test Data of High-Speed Train

The defects data are collected from a pure rolling test platform of a high-speed train. The train moving mechanism consists of two groups of bogie systems. Each group of bogies has two groups of wheels and axle box bearings. The train was driven by a drive motor. In Figure 4, the left half is the test bench and the right half is the test sensor installation schematic. In this test, the bearing disassembled from the in-service train was used to simulate the failure and its structural parameters are shown in Table 2. The data were collected at a train speed of 100 km/h with a sampling frequency of 25,600 Hz.

Figure 5 shows the time domain waveforms of rolling bearings with failure-free state and different degrees of outer ring fault state. From the figure can be observed that with the deepening of the fault degree, the vibration amplitude of the signal was larger, but the difference between failure-free and moderate failure was small, difficult to distinguish.

Selecting 50 samples from failure-free bearings signal in the healthy state as the train dataset, the length of each group of samples was 10,240, and also collecting 50 samples from failure-free signal and three different fault degree signals as test dataset, and the data length was also 10,240. The train samples were input into the DRSN-LSTM reconstruction model, and the network loss value was constructed with the mean square error of the input signal and reconstructed signal.

As shown in Figure 6, the loss value of the network decreased with the increase of epochs, and the loss value near 200 iterations was close to zero, indicating that the model converged at this time. The input signal and reconstructed signal were extracted using multi-domain feature extraction to obtain the input signal feature extraction vector and reconstructed signal feature extraction vector, respectively; and the mean squared error values of the two vectors were calculated as shown in Figure 7. From Figure 7, it can be seen that there was a big difference between failure-free samples and samples with different degrees of failures, and the mean squared error indicator increased with the aggravation of failures.

## 6. Data Analysis of Rolling Bearing from XJTU-SY Test Stand

### 6.1. Description of Datasets

In this paper, the XJTU-SY rolling bearing accelerated full-life dataset has been used to verify the effectiveness of the proposed method in performance degradation assessment under actual working conditions. The test rig consisted of an AC motor, motor speed controller, support bearings, rotating shaft, acceleration sensor, test bearing and hydraulic pressurization system, etc., as shown in Figure 8. During the test, bearing data were collected under three operating conditions, each corresponding to five test bearings. In this paper, the vibration signal data of bearing 1 under the third operating condition was used for analysis. Vibration signals were collected at a sampling frequency of 25.6 kHz, speed of 2400 r/min, sampling interval of 1 min, and sampling time of 1.28 s. The total time of the test was 42 h 18 min, and a total of 2538 sets of data were collected. Finally, the bearing had an outer ring failure with a failure frequency of 123.3 Hz.

The time-domain waveform of the whole life-circle vibration signal is shown in Figure 9 and its six time-domain indicators are shown in Figure 10. As can be seen from Figure 10, the amplitude of the whole-life vibration signal was stable between −2 and 2 from the beginning to the 2300th group of samples. The amplitude started to rise sharply from the 2300th to the last group of samples. As can be seen from Figure 11, only the root mean square value indicator was similar to the fluctuation trend of signal amplitude in Figure 10. The cliff value and waveform factor indicators have a small fluctuation in the early stage, and then decreased and increased in the later stage, which was not very similar to the fault development trend, the peak factor, pulse factor and margin factor. The indicators have a large fluctuation in the early stage, and also have the trend of decreasing and then increasing in the later stage, which cannot effectively detect the early fault.

### 6.2. Performance Degradation Assessment Analysis

The developed model was implemented using Pytorch-1.10.0. Experiments were conducted on a laptop with an i7-11800H CPU and a NVIDIA RTX3060 GPU. The optimal hyperparameters were finally determined by changing them several times to debug the program. So, the iterations of network (Epoch) were 300, the number of batches (Batch Size) was 2, and the learning rate was 0.001. The gradient descent optimizer for network training was Adam, and the loss function was the mean square error loss function.

The first 400 sets of health state data of the bearing fatigue test full life bearing 1 were input into the model for training. The length of each data was intercepted L = 10,240. Four hundred sets of data were disrupted for each training to prevent the model from overfitting due to sample sorting. The network loss value was constructed using the mean square error between the input signal and the reconstructed signal. As shown in Figure 11, the network loss value decreased with the increase in training times, which shows that the reconstructed model was optimized continuously. To prevent overfitting and control the loss value at about 0.05, the early stop method was used to end training after 300 iterations, at which time the reconstructed model only extracted the signal features with the health state and could reconstruct itself.

The bearing 1 full 0-life data, i.e., 2538 sets of data, were all input one by one into the trained DRSN-LSTM reconstruction model. The length of data was also 10,240. The input signal was reconstructed using the DRSN-LSTM reconstruction model to output the reconstructed signal. The input signal and reconstructed signal were extracted using multi-domain feature extraction to obtain the input signal feature extraction vector and reconstructed signal feature extraction vector, respectively. The mean square error between two vectors was calculated to obtain the bearing performance degradation assessment graph shown in Figure 12. In Figure 12, the solid line is the degradation curve of the performance of the fault degree in this paper, and the dotted line is the 3 σ adaptive alarm threshold. Due to the presence of noise interference and the asymptotic nature of the fault degree, the resulting mean square error values were smoothed using five points, i.e., the mean square error values obtained for each sample in Figure 12 were obtained using linear fitting of itself and two adjacent points.

As can be observed from Figure 12, the basic trend of the degradation curve is rising, which is consistent with the evolution process of the rolling bearing failure degree. Compared with the degradation plot of the six time-domain indicators in Figure 12, this indicator can effectively remove the effect of noise and is more sensitive to early failures. Analysis of Figure 12 shows that the performance degradation assessment of XJTU-SY rolling bearing 1 with the DRSN-LSTM reconstruction model can be roughly divided into four stages, i.e., healthy stage, early failure stage, moderate failure stage and extreme deterioration to failure stage.

In Figure 12, testing data from the 415th sample to its next three samples mean square error indicators are above the adaptive warning threshold. Thus, it can be considered that the bearing early failure has occurred in the 415th sample The bearing was in the healthy stage from the 1st sample to the 415th sample. In the 415th to the 767th samples, the mean square error indicator of the bearing showed an overall increasing trend. It indicates that in this process the early failure of the bearing was accumulating and the deterioration was aggravating.

The mean square error indicator dropped slightly from the 767th sample to the 808th sample, which indicated that, previously, the early failure of the bearing appeared during a short period of grinding flat. From the 808th to the 2046th sample, the mean square error indicator generally showed an increasing trend, which indicated that at this time the bearing is in the moderate failure stage. From the 2046th to the 2403rd sample, the mean square error indicator was generally decreasing, and the bearing was worn out again at this stage. During the period from sample No. 415 to sample No. 2403, the trend of the mean square error indicator of testing data increased, then decreased, and then increased and decreased again, which indicated that the bearing to be tested in its stage had the phenomenon of “gradual aggravation of failure—failure point wears out—deterioration again—deterioration wears out”. In the 2403rd to 2538th samples, the mean square error indicator degradation curve showed an upward trend in general, although there was a certain fluctuation in the middle stage, which indicated that the bearing had occurred the phenomenon of “wears out—deterioration” again. Because the time of its multiple fluctuation phenomenon was greatly shortened, it indicated that the bearing was in the stage of extreme deterioration to failure.

## 7. Comparison of Methods and Validation

To verify the superiority of the proposed method, the performance degradation of the same bearing data is assessed using multiple comparison methods under the same model hyperparameters, and the result curves are shown in Figure 13. To verify that the model combination of DRSN-LSTM can effectively preserve the temporal characteristics of signal while extracting features, the signal reconstruction models are built with CNN and LSTM, respectively, the performance degradation indicator is constructed with multi-domain feature extraction. The curves are shown in Figure 13a,b, respectively. From Figure 13a, it can be seen that the performance degradation indicator fluctuates between 0 and 0.1 from the first sample to the 2330th sample, and the degradation indicator of the signal only exceeds the alert line at the 2351st sample. In Figure 13b, the assessment curve gradually rises slowly from the first sample to the 1960th sample, and the performance degradation indicator only exceeds the alert line in the 1961st sample, indicating that the performance degradation assessment using the CNN method or LSTM is not advantageous.

In order to study the influence of DRSN model on bearing performance degradation assessment results, the CNN-LSTM model was used to build the signal reconstruction model. The mean square error between reconstructed signal and input signal after multi-domain feature extraction was used as the performance degradation indicator. The result curves are shown in Figure 14c. The curves in the first sample to the 2400th sample mean square error slowly increased, and it only exceeds the alert line in the 2413th sample. Thus, CNN-LSTM cannot effectively detect early bearing faults, which indirectly illustrates the effectiveness and superiority of DRSN for signal reconstruction.

The DRSN-LSTM model was used to reconstruct the signal, in the model testing stage, without feature extraction, the mean square error between reconstructed signal and input signal was directly used as the performance degradation indicator. The curve shown in Figure 13d from the first sample to the 2340th sample is nearly horizontal, and the mean square error does not exceed the alert line until the 2346th sample, which is not advantageous as the performance degradation indicator compared to reconstructed signal and input signal after multi-domain feature extraction. Thus, it shows the importance of using multi-domain feature extraction in this paper.

In this paper, the early failure points detected for bearing 1 are 540 points ahead of the 955th sample early failure point detected using the multivariate state estimation model in the literature [21]. This is 232 sample points ahead of the 647th sample early failure point detected with the AE-KSVD network model in Chapter 5 of the literature [7]. It proves that the deep learning model has a great advantage over the mathematical model and the shallow network model.

To further test the feasibility of the DRSN-LSTM reconstruction model, the envelope spectrum spectral peak factor was used as the optimization indicator for adaptive resonance demodulation of the 414th and 415th sample signals. The envelope spectrum analysis was implemented after resonance demodulation, and the results are shown in Figure 14 and Figure 15. It can be seen that the noise component of the signal was significantly weakened after resonance demodulation. The cyclic pulse of signal can be observed in Figure 14c, and the frequency 124.6 Hz and its corresponding double frequency 242.3 Hz which was similar to the characteristic frequency of the outer ring appear in the spectrum Figure 14d. Moreover, there was no component similar to the frequency and double frequency of the characteristic of the outer ring in Figure 15. The optimal filtering band of the 415th sample signal has a great difference from that of the 414th sample signal. It can be confirmed that an early outer ring failure occurred in the bearing at this time, which confirms the effectiveness of the proposed method.

## 8. Conclusions

Based on the advantages of the deep learning model, a rolling bearing performance degradation assessment method incorporating noise reduction features and timing information is proposed to solve the problem that the input signal of the traditional performance degradation assessment model is mixed with a large amount of noise and transmission path interference, and its timing information is usually ignored when the model processes the input signal, which leads to an unsatisfactory bearing performance assessment. The fatigue test data show that.

(1)The reconstruction model of rolling bearing performance degradation assessment proposed in this paper removes the noise in the signal while retaining its time information when reconstructing the signal, so as to better reflects the useful components of the original signal and improve the validity of the performance degradation assessment results.(2)The reconstruction model for rolling bearing performance degradation assessment proposed in this paper is an end-to-end data-driven model, in which the model automatically learns spatial and temporal features in the signal. The tedious feature engineering processing and expert knowledge experience of traditional methods are not required.(3)The proposed method is more sensitive to early faults than mathematical models, shallow networks or other deep learning models, more similar to bearing fault trends, and has the advantages of better consistency and monotonicity.

## Figures and Tables

**Figure 1 sensors-23-05910-f001:**
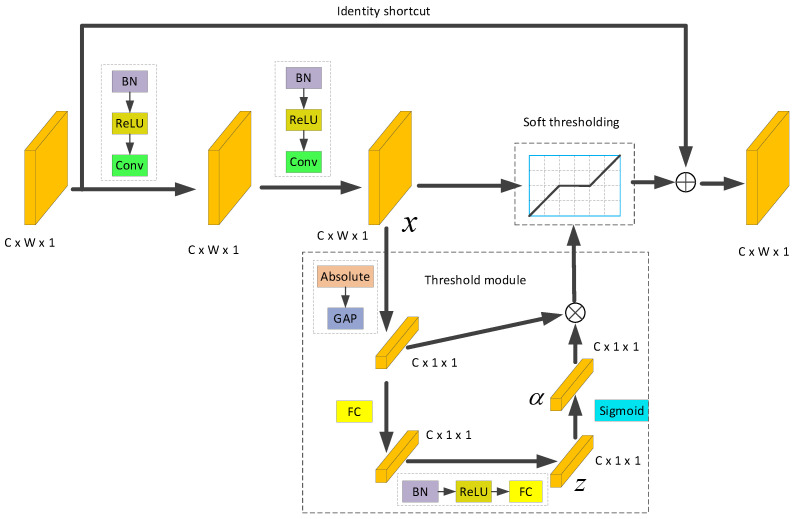
Residual shrinkage building unit (RSBU).

**Figure 2 sensors-23-05910-f002:**
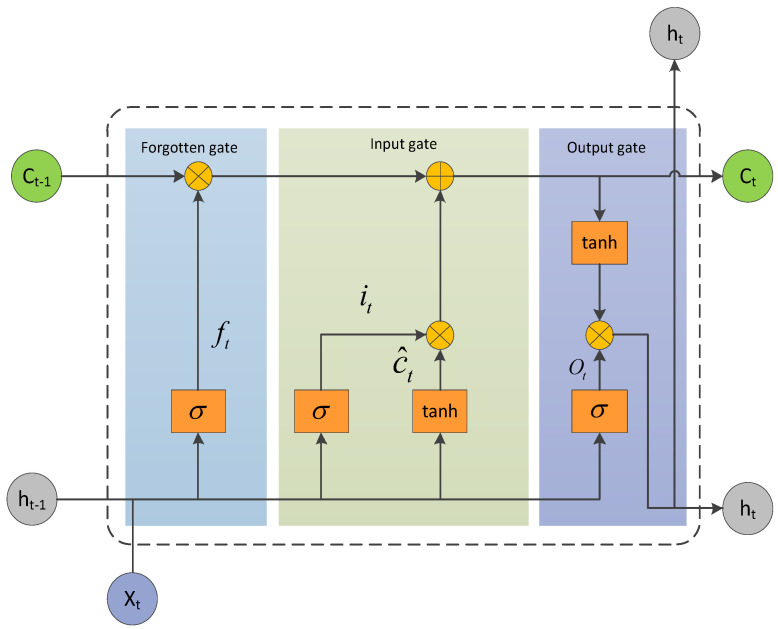
The internal structure diagram of LSTM.

**Figure 3 sensors-23-05910-f003:**
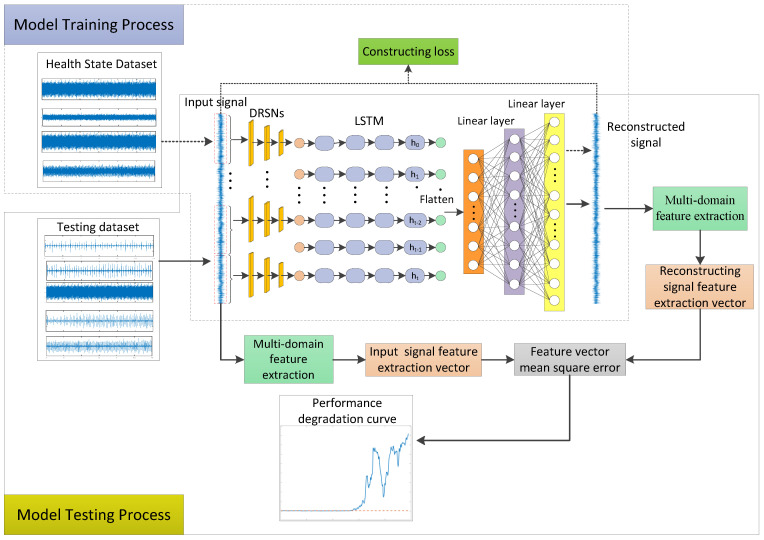
Procedure of bearing performance degradation assessment.

**Figure 4 sensors-23-05910-f004:**
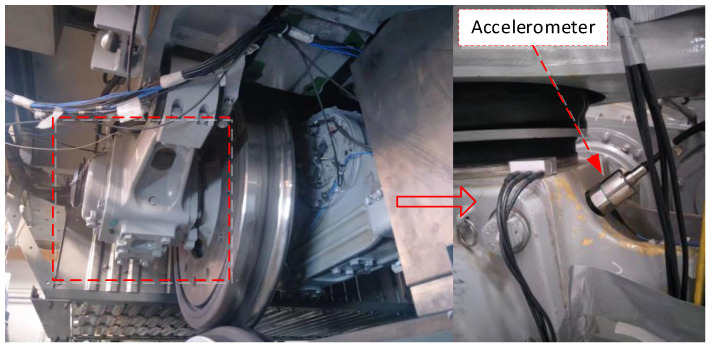
Rolling test rig for axle box bearings of high-speed trains.

**Figure 5 sensors-23-05910-f005:**
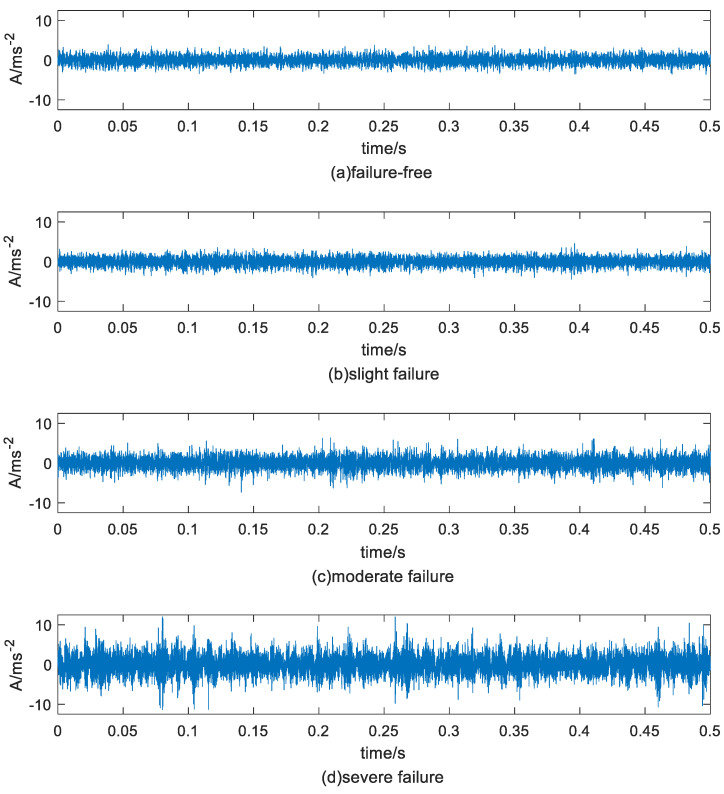
Signals from different bearing conditions.

**Figure 6 sensors-23-05910-f006:**
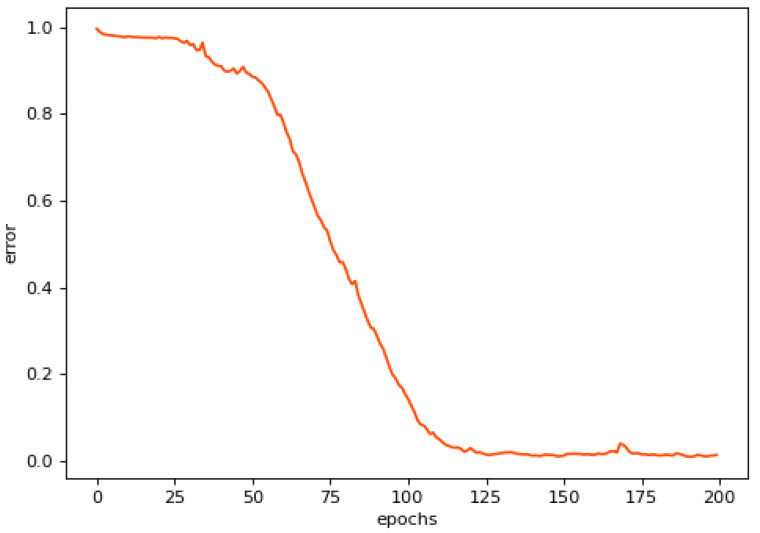
Training error of Axle Box Bearing.

**Figure 7 sensors-23-05910-f007:**
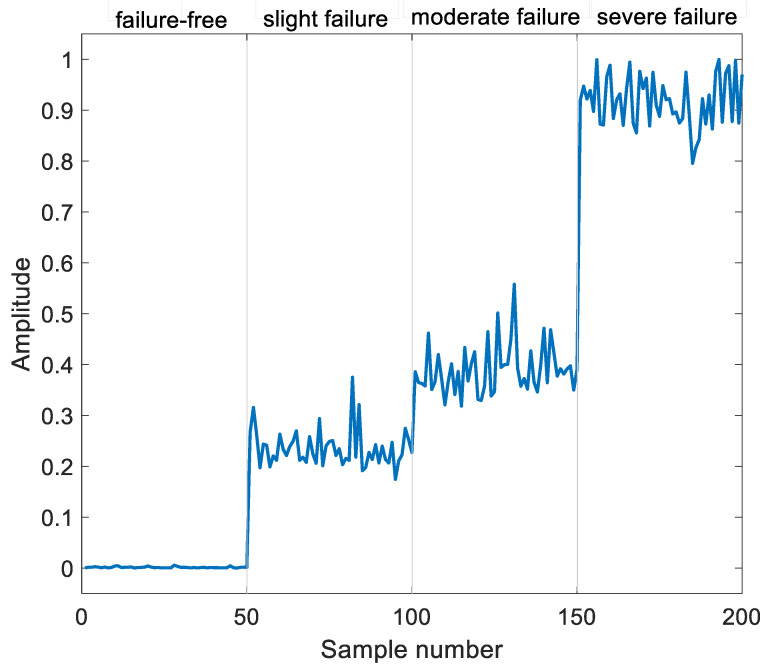
Mean square error indicator from different bearing conditions.

**Figure 8 sensors-23-05910-f008:**
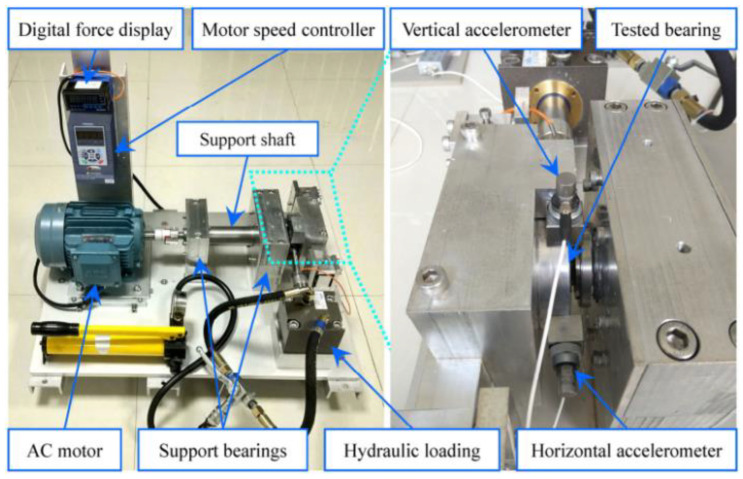
Test rig of Bearing with accelerated life.

**Figure 9 sensors-23-05910-f009:**
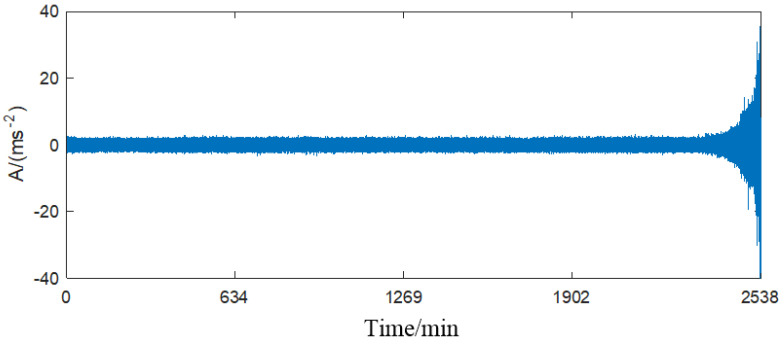
Time-domain waveform diagram of bearing life-circle vibration signal.

**Figure 10 sensors-23-05910-f010:**
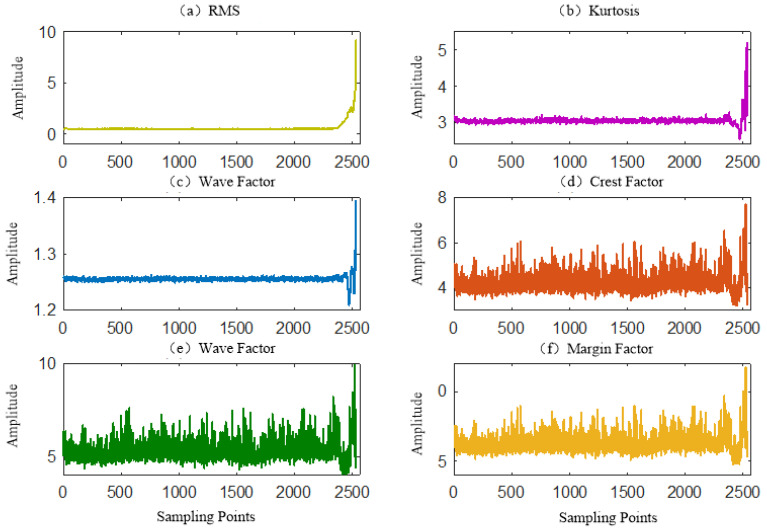
Time indicator of bearing.

**Figure 11 sensors-23-05910-f011:**
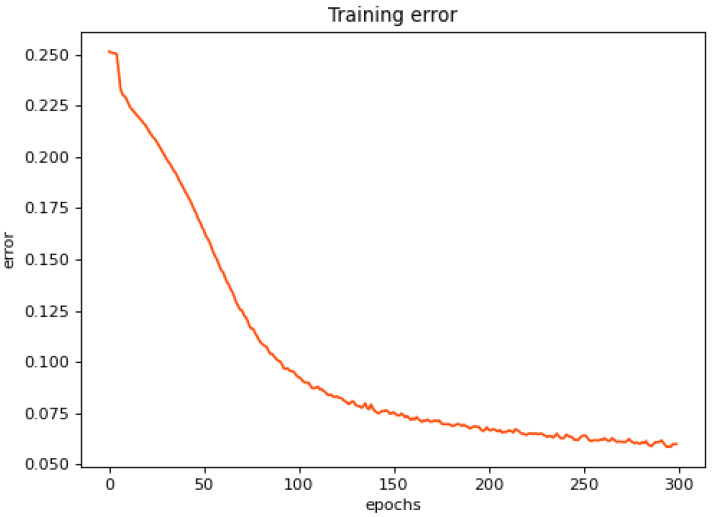
Training error of Rolling Bearing from XJTU-SY Test Stand.

**Figure 12 sensors-23-05910-f012:**
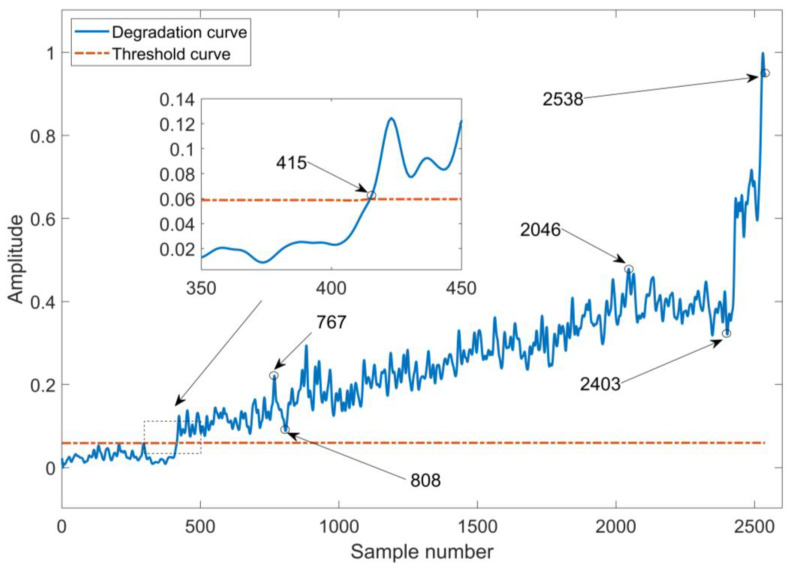
PDA results of bearing based on the proposed method.

**Figure 13 sensors-23-05910-f013:**
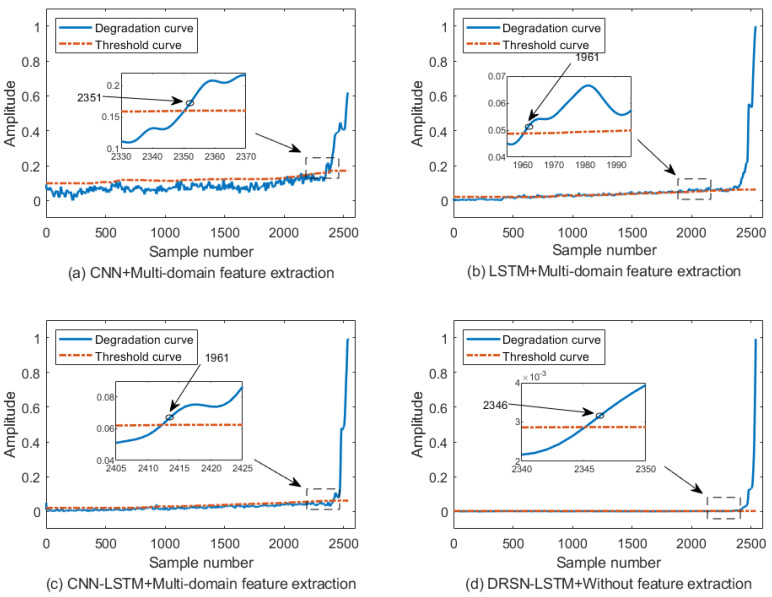
PDA results of bearing based on comparative method.

**Figure 14 sensors-23-05910-f014:**
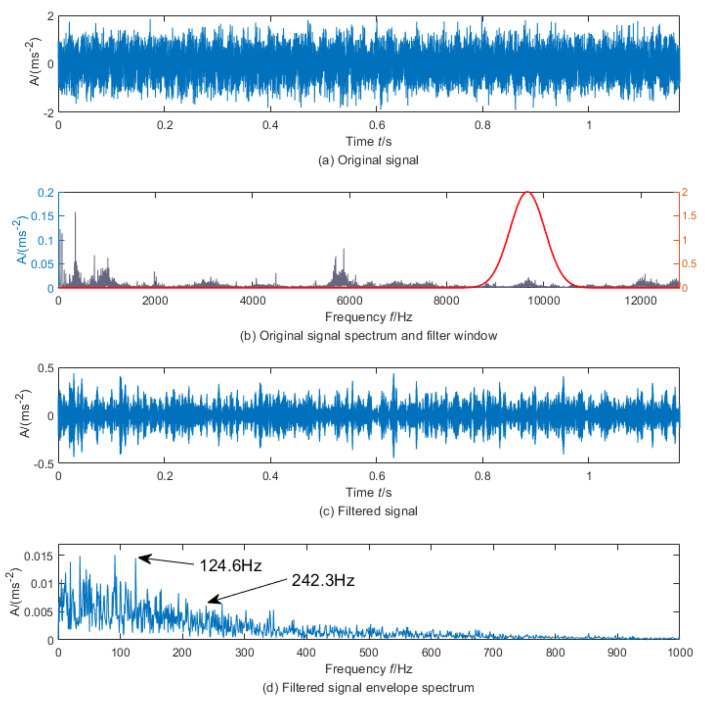
Envelope demodulation of No. 415 samples.

**Figure 15 sensors-23-05910-f015:**
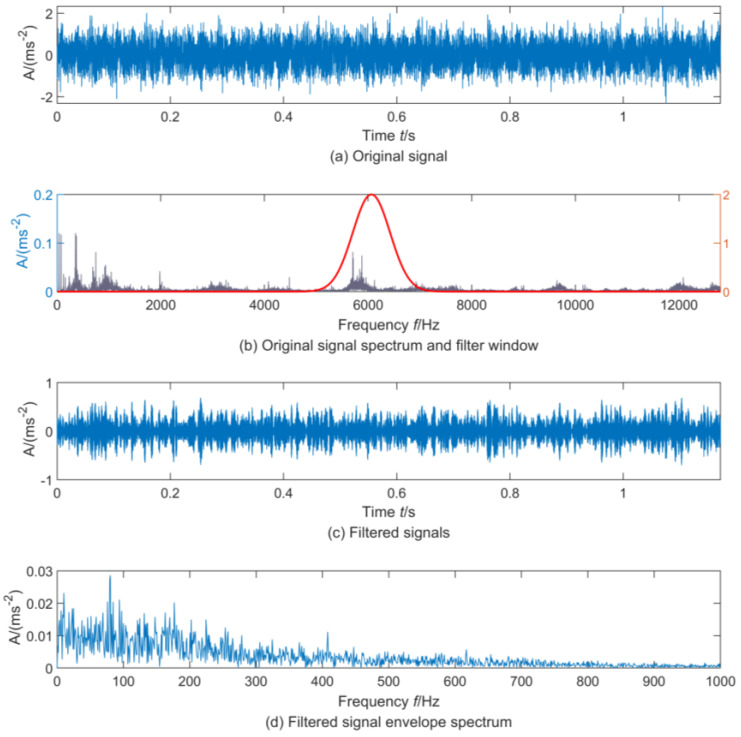
Envelope demodulation of No. 414 sample.

**Table 1 sensors-23-05910-t001:** Structure parameters of the DRSN-LSTM reconstruction model.

	Networks	Layers	Kernel Size	Stride	Padding	Input	Output
Encoder	20 × DRSNs	Conv, BN, ReLU	3 × 1	1	1	1 × L	64 × L
		Block1		1		64 × L	64 × L
		Block2		2		64 × L	128 × L/2
		Block3		2		128 × L/2	256 × L/4
		Block4		2		256 × L/4	1 × L/8
	LSTM					20 × L/8	20 × L/32
Decoder		Linear1				1 × 5L/8	5L/2
		Linear2				5L/2	1 × 20L

**Table 2 sensors-23-05910-t002:** Structure parameters of rolling bearings.

Radius of Section Circle D/mm	Rolling Body Diameter d/mm	Number of Rolling Elements *n*	Contact Angle α/(°)
183.929	26	19	10

## Data Availability

The data used to support the finding of this study are available from the corresponding author upon request.

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
