# Peer review of "Performance Degradation Assessment of Railway Axle Box Bearing Based on Combination of Denoising Features and Time Series Information"

_sensors, 2023, doi:10.3390/s23135910_

Round 1

Reviewer 1 Report

1.     The quantitative results should also be incorporated in the abstract and conclusion section.

2.     What threshold value have been considered in this research work?

3.     How authors have optimized the hyperparameters of the LSTM?

4.     How authors have dealt with underfitting and overfitting of the data?

5.     Please cite literatures from the year 2022 and 2023. Some of suggestion are as follows

·       An effective health indicator for bearing using corrected conditional entropy through diversity-driven multi-parent evolutionary algorithm

·       Bearing defect identification via evolutionary algorithm with adaptive wavelet mutation strategy

·       Unsupervised Learning Model of Sparse Filtering Enhanced Using Wasserstein Distance for Intelligent Fault Diagnosis

No Comments

Reviewer 2 Report

The paper deals with the Performance Degradation Assessment of Railway Axle Box Bearing Based on Combination of Denoising Features and Time Series Information.

Even if the topic is relevant, this reviewer does not suggest the publication of this manuscript on this journal because it is not clear the novelty.

Rolling Bearing Performance Degradation is a well-known topic.

https://doi.org/10.3390/machines10080677

https://doi.org/10.3390/s23031110

https://doi.org/10.3390/sym11081064

If the authors have found a novel approach to estimate the performance degradation, it must be compared with existing methods (complexity, time consumption, generality). Otherwise, it remains a simple exercise.

Moreover:

Fig. 8 has been found in several papers.

Fig. 15 reports titles and axis in Chinese.

Reference [21] cannot be found in the literature. LIU Z G, XIONG G L, ZHANG L. MSET Model of Rolling Bearing Performance Degradation and Its Fault Warning [J]. Machine Design and Research. 2021, 37(02): 60-65.

English must be revised

Author Response

Point 1: Fig. 8 has been found in several papers.

Response 1: XJTU-SY rolling bearing accelerated full-life dataset is a well-known dataset to assess Rolling Bearing Performance Degradation model. For objectivity and authenticity, we chose this dataset to verify the superiority of proposed model.

Point 2: Fig. 15 reports titles and axis in Chinese.

Response 2: Fig. 15 reports titles and axis have modified to English.(p17)

Point 3: Reference [21] cannot be found in the literature. LIU Z G, XIONG G L, ZHANG L. MSET Model of Rolling Bearing Performance Degradation and Its Fault Warning [J]. Machine Design and Research. 2021, 37(02): 60-65.

Response 3: Due to the addition of three new references, the original reference [21] was changed to new reference [24], and the reference was replaced with another research content. The reference have been added below, please check.

[24]         Wang B, Lei Y, Li N, et al. A Hybrid Prognostics Approach for Estimating Remaining Useful Life of Rolling Element Bearings[J]. IEEE Transactions on Reliability. 2018: 1-12.

Round 2

Reviewer 1 Report

The manuscript in the correct form can be accepted for publication.

Reviewer 2 Report

The Authors did not answer to the request "

Rolling Bearing Performance Degradation is a well-known topic.

https://doi.org/10.3390/machines10080677

https://doi.org/10.3390/s23031110

https://doi.org/10.3390/sym11081064

If the authors have found a novel approach to estimate the performance degradation, it must be compared with existing methods (complexity, time consumption, generality). Otherwise, it remains a simple exercise."

Moreover, the overall quality of the manuscript has not be improved.

For this reason a revision is needed.

Minor issues have to be solved.
